# UPS: OPTIMIZING UNDIRECTED POSITIVE SPARSE GRAPH FOR NEURAL GRAPH FILTERING

## ABSTRACT

In this work we propose a novel approach for learning graph representation of the data using gradients obtained via backpropagation. Next we build a neural network architecture compatible with our optimization approach and motivated by graph filtering in the vertex domain. We demonstrate that the learned graph has richer structure than often used nearest neighbors graphs constructed based on features similarity. Our experiments demonstrate that we can improve prediction quality for several convolution on graphs architectures, while others appeared to be insensitive to the input graph.

## 1 INTRODUCTION

Recently we have seen a rise in deep learning models, which can account for non-linearities and fit a wide range of functions. Multilayer perceptron (MLP), a general purpose neural network, is a powerful predictor, but requires too many parameters to be estimated and often faces the problem of over-fitting, i.e. learns to almost exactly match training data and unable to generalize when it comes to testing.

While MLPs treat all features equally, which partially is the cause of excessive number of parameters, Convolutional Neural Networks (CNNs) have significantly fewer parameters and demonstrate groundbreaking results when it comes to object recognition in images (Krizhevsky et al., 2012). The parameter reduction is due to utilizing convolutional operation: a window is sliding through the image and applying same linear transformation of the pixels. The number of parameters then is proportional to the size of the window rather than polynomial of the number of data features as in the case of the MLPs.

Indeed images posses a specific structure, which can be encoded as a lattice graph, that makes the sliding window procedure meaningful, but inapplicable outside of the image domain. In recent years there have been multiple works (cf. Bronstein et al. (2017) for an overview) on generalizing convolution operation to a general domain, where graph is not a lattice. Citing Defferrard et al. (2016) - "classification performance critically depends on the quality of the graph", nonetheless the problem of learning the graph useful for prediction has not been addressed so far and the graph was either known or pre-estimated only based on feature similarity in all of the prior work.

There are two major challenges when estimating the graph inside the neural network architecture. First is the architecture itself - majority of the neural networks rely on gradient optimization methods, but the graph is often used in such ways that it is not possible to obtain its gradient. In Section 3 we define a novel neural network architecture which is differentiable with respect to the graph adjacency matrix and built upon graph filtering in the vertex domain, extending the linear polynomial filters of Sandryhaila & Moura (2013). Second problem is the series of constraints that are often imposed on the graph and therefore its adjacency. In Section 2 we show how the three common graph properties, undirected sparse edges with positive weights, can be enforced by only utilizing the gradient obtained through backpropagation, therefore allowing us to utilize any of the modern deep learning libraries for graph estimation. In Section 4 we discuss other graph based neural networks and evaluate them from the perspective of graph estimation. In Section 5 we analyze graph estimation and interpretation for text categorization and time series forecasting. We conclude with a discussion in Section 6

## 2    GRAPH OPTIMIZATION BASED ON BACKPROPAGATION

In this section we provide an optimization procedure for learning adjacency matrix of a graph with various properties of interest, assuming that we can obtain its derivative via backpropagation. In a subsequent section we will present novel neural network architecture that will allow us to get the derivative and utilize the graph in meaningful way.

Let data $X \in \mathbb{R}^{N \times D}$ with $N$ observation, $D$ features and response $Y \in \mathbb{R}$ (or $Y \in \mathbb{N}$ for classification). Graph $G$ among data features can be encoded as its adjacency matrix $A \in \mathbb{R}^{D \times D}$. Our goal is to estimate function $\hat{Y} := f_W(X, A)$, where $W$ are weight parameters, that minimize some loss $L := L(\hat{Y}, Y)$. We assume that we are able to evaluate partial derivative $\frac{\partial L}{\partial A}$. In the most general case, when edges of $G$ can be directed, have negative weights and $G$ can be fully connected, we perform the update $A := A - \gamma \mathcal{G}\left(\frac{\partial L}{\partial A}\right)$, where $\mathcal{G}(\cdot)$ depends on the optimizer (e.g., identity function for vanilla gradient descent) and $\gamma$ is the step size. Nonetheless, in the majority of the applications, $G$ is desired to have some (or all) of the following properties:

- **U**ndirected graph, in which case $A$ is restricted to be symmetric.
- Have **P**ositive edge weights, in which case $A \in \mathbb{R}_+^{D \times D}$.
- Be **S**parsely connected, in which case $A$ should contain small proportion of non-zero entries.

First two properties are necessary for the existence of the graph Laplacian, crucial for the vast amount of neural networks on graphs architectures (e.g., Bruna et al. (2013); Henaff et al. (2015); Defferrard et al. (2016)). Third property greatly reduces computational complexity, helps to avoid overfitting and improves interpretability of the learned graph. We proceed to present the Undirected Positive Sparse $\mathcal{UPS}$ optimizer, that can deliver each of the three properties and can be easily implemented as part of modern deep learning libraries.

**Remark**    When node classification is of interest, our approach can be applied to graph between observations (e.g. social networks), then $A \in \mathbb{R}^{N \times N}$.

### 2.1    UNDIRECTED GRAPH

When $G$ is desired to be undirected, its adjacency $A$ is a symmetric matrix, hence $A_{ij}$ and $A_{ji}$ are the same parameters. When backprop is used for gradient computation, this fact is not accounted for, but can be adjusted via the gradient correction

$$\mathcal{U}\left(\frac{\partial L}{\partial A}\right)_{ij} = \frac{\partial L}{\partial A_{ij}} + \mathcal{I}(i \neq j)\frac{\partial L}{\partial A_{ji}}, \text{ for } i, j = 1, \ldots, D. \tag{1}$$

Correctness of this procedure can be easily verified. Note that for modern stochastic optimization methods (e.g., Adam (Kingma & Ba, 2014)) the corrected gradient $\mathcal{U}\left(\frac{\partial L}{\partial A}\right)$ should be used for moment computations.

### 2.2    POSITIVE WEIGHTS

Restricting edge weights of the graph to be positive is necessary for the existence of the graph Laplacian and can help with interpretability of the resulting graph. To achieve positive weights we need to add an inequality constraint of the form $A_{ij} \geq 0$ for $i, j = 1, \ldots, D$ to our optimization task. Constrained optimization has been widely studied and multiple techniques are available. Given that we are building our optimization on top of the backprop, the most natural solution is the projected gradient descent. This method has been previously shown to be effective even in the non-convex setups (e.g., Nonnegative Matrix Factorization (Lin, 2007)). Projected gradient for positive weights constraint acts as follows:

$$A := \mathcal{P}\left(A - \gamma \mathcal{G}\left(\mathcal{U}\left(\frac{\partial L}{\partial A}\right)\right)\right), \text{ where } \mathcal{P}(x) = \mathcal{I}(x \geq 0)x. \tag{2}$$

Projection operator $\mathcal{P}$ is applied elementwise. Another option is to consider adding a barrier function (i.e. elementwise logarithm of $A$) to the objective function, but we found projected gradient to be simpler and better aligning with out next step.

### 2.3 SPARSITY

Sparsity is perhaps the most crucial property for several reasons. In modern high dimensional problems, it is almost never the case that graph is fully connected, hence adjacency $A$ should contain some zero entries. Sparsity greatly reduces computational complexity of any neural network relying on the graph, especially when graph optimization is considered as in our work. Finally, sparse graphs are much more interpretable. Variable selection is an active research area in a variety of domains (cf. Fan & Lv (2010)) and one of the dominant approaches is due to the $L_1$ penalty on the object that is desired to be sparse, in our case penalty is $g_\lambda(A) = \lambda \sum_{i,j} |A_{ij}|$, which is combined with the loss $L(X, A, W)$ to form a new objective function. It is known that $L_1$ norm is not differentiable at 0, although, similar to gradient, subgradient descent optimization can be used. They key disadvantage of such approach is that it does not actually obtain sparse solution. Instead we propose to use proximal gradient method (cf. Section 4 of Parikh et al. (2014)), which again aligns well with backprop based optimization. Proximal operator of the $L_1$ penalty $g_\lambda(A)$ is the soft thresholding operation:

$$\text{prox}_{g_\lambda}(x) := \mathcal{S}_\lambda(x) = (x - \lambda)_+ - (-\lambda - x)_+, \text{ where } x_+ := \max(0, x). \tag{3}$$

Then our final, sparsifying, step is:

$$A := \mathcal{S}_{\gamma\lambda}\left(\mathcal{P}\left(A - \gamma\mathcal{G}\left(\mathcal{U}\left(\frac{\partial L}{\partial A}\right)\right)\right)\right). \tag{4}$$

Notice that we threshold by $\lambda$ scaled by the step size $\gamma$ and that soft thresholding operation can be simplified for positive weights $\mathcal{S}_{\gamma\lambda}(x) = (x - \gamma\lambda)_+$.

**Remark** Another graph property that is sometimes of interest is the presence of self connections. If one wants to prohibit self connections, it can trivially be done by setting the diagonal of $A$ to 0 and not performing any updates on the diagonal. We do not enforce this since $\mathcal{UPS}$ can estimate what self connections, if any, should be present in the graph via the proximal 4 step.

## 3 GRAPH BASED NEURAL NETWORK ARCHITECTURES

The key assumption of the $\mathcal{UPS}$ optimizer is that we can evaluate the partial gradient of the adjacency matrix. We propose a novel neural network architecture arising from the Graph Signal Processing (GSP) literature that satisfies the assumption.

### 3.1 GRAPH FILTERING

A prominent way to improve the quality of features of $X \in \mathbb{R}^D$ (we consider a single observation in this section to simplify the notations) is to process it as a signal on graph among data features with adjacency matrix $A \in \mathbb{R}^{D \times D}$. GSP (see Shuman et al. (2013) for an overview) then allows us to do filtering in the vertex domain as follows:

$$f^{(k)}(X_i) = \sum_{j \in \mathcal{N}(i,k)} b_{i,j}^{(k)} X_j, \tag{5}$$

where $j \in \mathcal{N}(i, k)$ iff $A_{ij}^k \neq 0$ and $b_{i,j}^{(k)}$ for $i, j = 1, \ldots, D$ are filtering coefficients. Equation 5 modifies signal at the $i$th feature by taking into account signals at features reachable in exactly $k$ steps. By varying $k = 1, \ldots, K$ we can extract new features that account for the graph structure in the data and combine them into filtered graph signal $f(X, A) \in \mathcal{R}^D$ used for prediction.

Sandryhaila & Moura (2013) proposed linear polynomial graph filter of the form:

$$H(A) = w_0 I_D + w_1 A + \ldots + w_K A^K, \quad b_0, \ldots, b_K \in \mathbb{R}. \tag{6}$$

Then the filtered signal is obtained via matrix multiplication $f(X, A) = XH(A)$ and is a special case of filtering in the vertex domain:

$$f(X_i, A) = \sum_{k=0}^{K} w_k f^{(k)}(X_i) = \sum_{k=0}^{K} w_k \sum_{j \in \mathcal{N}(i,k)} (A^k)_{ij} X_j \text{ for } i = 1, \ldots, D \text{ and } A^0 := I_D. \tag{7}$$

Filtering in Eq. 7 can be used for graph optimization with $\mathcal{UPS}$, but it possesses several limitations. Filtered signals $f^{(k)}(X_i)$ are combined in linear way and choice of filtering coefficients $b_{i,j}^{(k)} = (A^k)_{ij}$ might lack flexibility.

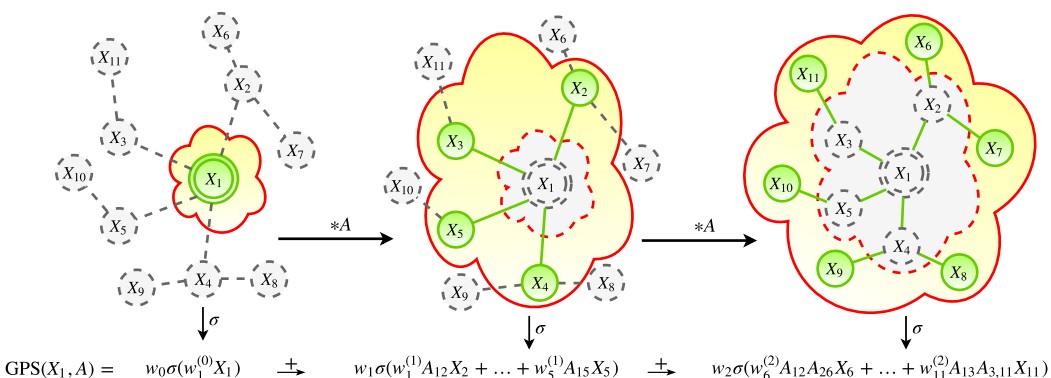

$$\text{GPS}(X_1, A) = \quad w_0\sigma(w_1^{(0)}X_1) \quad \underset{+}{\pm} \quad w_1\sigma(w_1^{(1)}A_{12}X_2 + \dots + w_5^{(1)}A_{15}X_5) \quad \underset{+}{\pm} \quad w_2\sigma(w_6^{(2)}A_{12}A_{26}X_6 + \dots + w_{11}^{(2)}A_{13}A_{3,11}X_{11})$$

Figure 1: Example of GPS of a vertex $X_1$. Dashed grey circles and lines are the inactive vertices and edges. Green solid circles and lines are the active ones. Note that vertices inside dashed red cloud can be active if graph has self connections — a decision made by $\mathcal{UPS}$.

### 3.2 GRAPH POLYNOMIAL SIGNAL NEURAL NETWORK

Neural networks are known to be much more effective than linear models and hence we address the two shortcomings of graph filter 7 with the following Graph Polynomial Signal (GPS) neural network:

$$\text{GPS}(X_i, A) = \sum_{k=0}^{K} w_k\sigma(f^{(k)}(X_i)) = \sum_{k=0}^{K} w_k\sigma\left(\sum_{j\in\mathcal{N}(i,k)} w_j^{(k)}(A^k)_{ij}X_j\right). \tag{8}$$

GPS directly utilizes filtered signals based on vertex neighborhoods of varying degree $k$ and allows for non-linear feature mappings. GPS example is given in Fig. 1. Last step is to build a mapping from the GPS features into the response $Y$. This can be done via linear fully connected layer or a more complex neural network can be built on top of the GPS features. Our architecture can be easily implemented using modern deep learning libraries and backprop can be used to obtain the partial derivative of the adjacency $A$, required by the $\mathcal{UPS}$ optimizer.

Role of weights $w_j^{(k)}$, $j = 1, \dots, D$, $k = 1, \dots, K$ is two fold — firstly, they scale the graph adjacency, which is crucial for proximal optimization 4. For inducing sparsity in the adjacency $A$ ideally we would penalize number of nonzero elements ($L_0$ norm) in $A$, but such penalty is known to be NP-hard for optimization, hence the $L_1$ is always used instead. The drawback of this choice is that we penalize nonzero edge weights $A_{ij}$ by their magnitude $|A_{ij}|$ which might be detrimental for the prediction. To avoid disagreement between $L_1$ penalty term and prediction quality, re-scaling with weights is helpful.

Second role of weights has the nature of weight sharing of classical CNN on images. For image data, objects are often considered to be location invariant, hence CNN shares same set of weights across the whole image. For a general data type considered in our work, there is no reason to make location invariance assumption. Instead we assume that weights should be shared among neighboring graph regions. In particular, observe that $\mathcal{UPS}$ can decide to partition the graph into multiple connected components, then GPS will enforce weight sharing inside each component by construction.

### 3.3 OTHER GRAPH ADJACENCY BASED NEURAL NETWORKS

GPS was designed to perform graph filtering based on its adjacency matrix in a way, that $\mathcal{UPS}$ optimizer can be used to learn the adjacency. There are few other architectures that require a graph be given, but can be combined with the $\mathcal{UPS}$ for graph learning. Scarselli et al. (2009) proposed Graph Neural Network (GNN) — a rather general framework for utilizing graph neighborhoods information. Some of the recent works on neural networks on graphs can be viewed as a special case of it (Bronstein et al., 2017). GNN does not utilize adjacency directly in the architecture, but

its modern variation Graph Convolutional Network (GCN) (Kipf & Welling, 2016) does so for the case of graph among observations. Their architecture can be modified for graph among features via stacking layers of the form:

$$X_i^{(k+1)} := \sigma \left( \sum_{j=1}^{D} w_j^{(k)} \tilde{A}_{ij} X_j^{(k)} \right) \text{ for } i = 1, \ldots, D, \quad (9)$$

where $X^{(0)} := X \in \mathbb{R}^D$ is an observation and $\tilde{A} = \mathcal{D}^{-\frac{1}{2}}(A + \mathcal{I}_D)\mathcal{D}^{-\frac{1}{2}}$ to enforce self connections, $\mathcal{D}_{ii} = 1 + \sum_j A_{ij}$ is a degree of vertex $i$. $w_j^{(k)} \in \mathbb{R}$ for $j = 1, \ldots, D, k = 0, \ldots, K - 1$ are the trainable parameters. Notice that since data in the applications we consider does not have multiple input channels, we modified the architecture to have different weights across the graph for every layer. Kipf & Welling (2016) show that this architecture does 1-hop filtering inside each layer. When multiple layers are stacked, resulting expression gets very complex due to non-linearities and can not be considered as filtering of higher degree in the sense of Graph Signal Processing as in Eq. 5. It is possible to use $\mathcal{UPS}$ with GCN for graph learning if we use $A$ instead of $\tilde{A}$ in Eq. 9, but the connection to graph filtering would be lost.

# 4 RELATED WORK

Deep learning on graphs has recently become an active area of research, but all of the prior work assumes the graph be given or estimated prior to model fitting, using, for example, kNN graph based on a feature similarity metric of choice. Henaff et al. (2015) formulated a supervised graph estimation procedure, but this again is done prior to their model fitting by training an MLP and utilizing first layer features for kNN graph construction. Another popular direction, motivated by the success of word embedding (Mikolov et al., 2013) in the NLP domain, is learning latent feature representations of the graph (Perozzi et al., 2014; Grover & Leskovec, 2016; Rossi et al., 2017). Here graph is also required as an input.

When it comes to architecture design involving graphs two approaches are usually distinguished — spatial and spectral (cf. Bronstein et al. (2017) for an overview). GPS is a spatial approach utilizing graph adjacency as building block in a suitable for graph optimization manner. We have already discussed two existing spatial approaches that can be combined with graph optimization. Next we discuss some other spatial and spectral approaches from the perspective of graph learning.

## 4.1 SPATIAL ARCHITECTURES

Niepert et al. (2016) proposed creating receptive fields using graph labeling and then applying a 1D convolution. Hechtlinger et al. (2017) suggested building CNN type architecture by considering neighborhoods of fixed degree using powers of the transition matrix. Graph node sequencing and neighborhood search are not differentiable and hence not compatible with the $\mathcal{UPS}$ optimization. Diffusion-Convolutional Neural Networks (DCNN) (Atwood & Towsley, 2016) use power series of the transition matrix and combine it with a set of trainable weights to do graph, node and edge classifications. Transition matrix requires positive weights and is restricted to be stochastic, which would complicate the optimization. DCNN pre-computes powers of the transition matrix and stores them as a tensor, which is not suitable for graph learning.

## 4.2 SPECTRAL ARCHITECTURES

Idea behind spectral architectures is to utilize eigenvector basis of the graph Laplacian to do filtering in the Fourier domain:

$$f(X_i) = \sum_{l=0}^{D-1} \langle X, u_l \rangle \hat{h}(\lambda_l) u_{li}, \quad (10)$$

where $u_l, \lambda_l \ l = 0, \ldots, D - 1$ are the eigenvectors and eigenvalues of the graph Laplacian $\mathcal{L}$ of $A$. Key choice one has to make when using spectral approach is the functional form of filter $\hat{h}(\lambda_l)$. Bruna et al. (2013); Henaff et al. (2015) proposed to use nonparametric filters $\hat{h}(\lambda_l) = w_l$, where $w_l, l = 0, \ldots, D - 1$ are trainable parameters. When graph learning comes into play, such

approach becomes inefficient since we would need to optimize for the eigenvectors of the graph Laplacian, which are not sparse even for sparse graphs and have to be constrained to be orthonormal. Additionally, they proposed to use hierarchical graph clustering for pooling, which one would have to redo on every iteration of the graph optimization as the graph changes.

Defferrard et al. (2016) utilized another filtering function $\hat{h}(\lambda_l) = \sum_{k=1}^{K} a^{(k)} \lambda_l^k$, which is appealing as it can be shown (Shuman et al., 2013) to perform filtering in the vertex domain 5 with filtering coefficients $b_{i,j}^{(k)} := a^{(k)} \mathcal{L}_{ij}^k$. They also utilized a Chebyshev polynomial approximation to the graph filter to bypass the necessity for computing the eigen decomposition of the Laplacian (Hammond et al., 2011). In the architecture design, they used graph coarsening and pooling based on node rearrangement, which, as discussed before, are non-differentiable operations. As in the case of transition matrix, optimizing for graph Laplacian would complicate the optimization, especially in the presence of Chebyshev polynomials.

## 5 EXPERIMENTS

In the experimental section our goal is to show that graph learned using $\mathcal{UPS}$ graph optimizer based on the GPS architecture can give additional insights about the data and can be utilized by other graph based neural networks.

### 5.1 EXPERIMENTAL SETUP

We fit the GPS architecture using $\mathcal{UPS}$ optimizer for varying degree of the neighborhood of the graph. Resulting graphs are then used to train ChebNet (Defferrard et al., 2016), ConvNet (Henaff et al., 2015), GCN (Kipf & Welling, 2016) as in Eq. 9 and Graph CNN (Hechtlinger et al., 2017). We also consider standard graph initialization using kNN graph, random graph and kNN graph based on the MLP features as in (Henaff et al., 2015) for all the above architectures and for the GPS *without* graph optimization. We used Adam (Kingma & Ba, 2014) for weight optimization and as a stochastic optimizer $\mathcal{G}$ for the $\mathcal{UPS}$ in Eq. 1, 2, 4.

### 5.2 TEXT CATEGORIZATION

In this experiment we provide thorough evaluation of various graph convolutional architectures using 20news groups data set. We keep the 1000 most common words, remove documents with less than 5 words and use bag-of-words as features. Training size is 9924 and testing is 6695. Results are presented in Table 1. We see that GPS with optimized graph can achieve good results, but fails when the graph is random or pre-estimated. Interestingly, ChebNet and ConvNet do not appear to be sensitive to the graph being used. This might be due to high number of trainable parameters in the respective architectures. GCN did poorly overall, but showed a relative improvement when the estimated graphs were used.

Next we compare the learned graph versus a kNN graph often used for initialization of various graph neural networks. We estimated the graph using $\mathcal{UPS}$ and $GPS_4$ architecture and used nested stochastic block model for visualization. Note that nested stochastic block model selects number of levels and blocks on each level of the hierarchy to minimize the description length of the graph (cf. Peixoto (2014a) and references therein). $GPS_4$ utilizes graph neighborhoods up to 4th degree and we see in Fig. 2a that there are 4 levels in between the input dimensions and the endpoint. Additionally note that intermediate levels have 18 and 5 blocks, which is roughly similar to 20 classes and 6 super classes (more general groupings of the news categories) of the 20news groups data, which is possibly due to the supervised nature of the graph. For comparison, we also provide a hierarchical structure of the 100NN graph in Fig. 2b, which is very poorly structured and only has two intermediate levels.

### 5.3 TIME SERIES FORECASTING

We use a dataset consisting of time series of visits to a popular website across 100 cities in the US. Visits counts were normalized by standard deviation. The task is to predict the average number of visits across cities for tomorrow. 3 years of daily data is used for training and testing is done on consecutive 200 days. Results are reported in Table 2. GPS demonstrated very good performance and

Table 1: Classification accuracy for different graphs and convolution architectures

|        | GPS 8 | ChebNet | ConvNet | GCN$_2$ Eq. 9 | Graph CNN |
|--------|-------|---------|---------|---------------|-----------|
| 100NN  | 26.14 | 56.77   | 56.77   | 36            | 56.34     |
| Random | 48.66 | 56.8    | 56.92   | 48.1          | 55.52     |
| MLP    | 49.63 | 57.15   | 56.91   | 46            | 56.86     |
| GPS$_1$ | 57.79 | 56.31  | 56.4    | 50.3          | 54.46     |
| GPS$_2$ | 57.1  | 56.98  | 56.55   | 49.6          | 57.1      |
| GPS$_3$ | 56.6  | 56.67  | 55.44   | 45.9          | 57.1      |

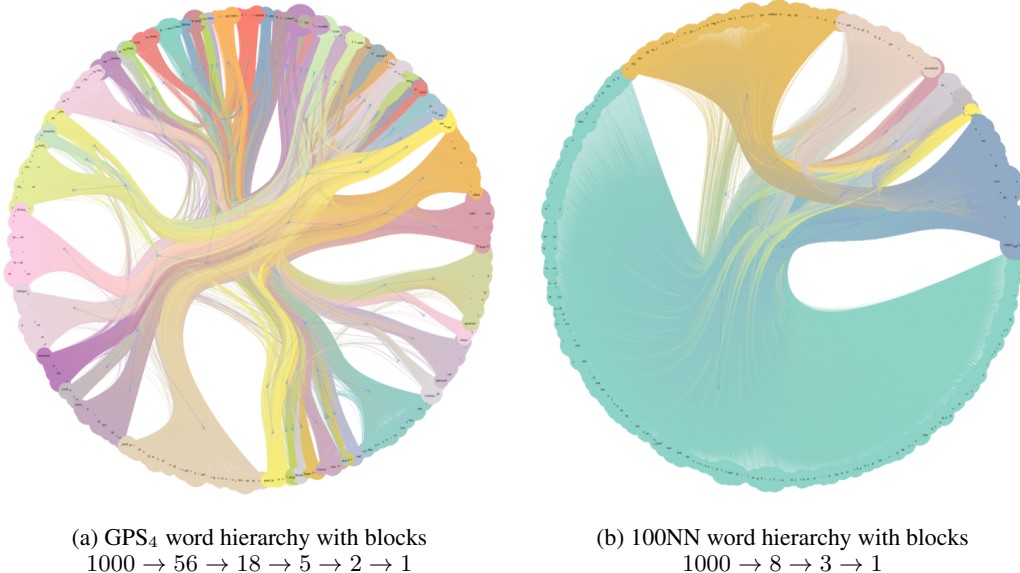

(a) GPS$_4$ word hierarchy with blocks
$1000 \rightarrow 56 \rightarrow 18 \rightarrow 5 \rightarrow 2 \rightarrow 1$

(b) 100NN word hierarchy with blocks
$1000 \rightarrow 8 \rightarrow 3 \rightarrow 1$

Figure 2: 20news hierarchical word relationships visualization using graph-tools (Peixoto, 2014b).

we can see noticeable improvement of the GCN$_3$ result when graph was learned for the neighborhood degree of 3 and higher. For ChebNet we report the best score instead of final one as it was overfitting severely. Nonetheless the best score appears to improve when the trained graph is used.

Table 2: MSE for different graphs and convolution architectures

|        | GPS 8 | ChebNet | ConvNet | GCN$_3$ 9 | Graph CNN |
|--------|-------|---------|---------|-----------|-----------|
| 10NN   | 0.27  | 0.31    | 0.32    | 0.276     | 0.41      |
| Random | 0.38  | 0.28    | 0.34    | 0.27      | 0.29      |
| MLP    | 0.251 | 0.27    | 0.29    | 0.226     | 0.32      |
| GPS$_2$ | 0.211 | 0.3    | 0.33    | 0.24      | 0.31      |
| GPS$_3$ | 0.204 | 0.26   | 0.34    | 0.19      | 0.32      |
| GPS$_4$ | 0.198 | 0.24   | 0.32    | 0.202     | 0.32      |

## 6 DISCUSSION

In this work, motivated by the rising attention to convolution on graphs neural networks, we developed a procedure and a novel architecture for graph estimation that can account for neighborhoods of varying degree in a graph. We showed that resulting graph has more structure then a commonly used kNN graph and demonstrated good performance of our architecture for text categorization and time series forecasting.

The worrisome observation is the insensitivity of some of the modern deep convolution networks on graphs to the graph being used. Out of the considered architectures, only GPS and GCN showed noticeable performance improvement when a better graph was used. These architectures stand out as they have much fewer trainable parameters and are more likely to suffer from a badly chosen graph. We think that a deep network utilizing the graph should not be able to produce any sensible result when the random graph is used.

When doing convolution on images, pooling is one of the important steps that helps to reduce the resolution of filters. It is unclear so far how to incorporate pooling into the GPS, while maintaining the ability to extract the gradient. This is one of the limitations of our approach and is of interest for further investigation.

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
