# OpenReview forum: "UPS: optimizing Undirected Positive Sparse graph for neural graph filtering"
_ICLR.cc/2018/Conference — Reject_

### Official Review · AnonReviewer2 · 2017-11-21
**Authors of this paper built a neural network architecture compatible with the novel approach for learning graph representation of the data, motivated by graph filtering in the vertex domain. The learned graph is demonstrated to be richer structures than nearest neighbor graphs.**

**Rating:** 6
**Confidence:** 3

**Review:**

Learning adjacency matrix of a graph with sparsely connected undirected graph with nonnegative edge weights is the goal of this paper. A projected sub-gradient descent algorithm is used. The UPS optimizer by itself is not new.

Graph Polynomial Signal (GPS) neural network is proposed to address two shortcomings of GSP using linear polynomial graph filter. First, a nonlinear function sigma in (8) is used, and second, weights are shared among neighbors of every data points. There are some concerns about this network that need to be clarified:
1. sigma is never clarified in the main context or experiments
2. the shared weights should be relevant to the ordering of neighbors, instead of the set of neighbors without ordering, in which case, the sharing looks random.
3. another explanation about the weights as the rescaling to matrix A needs to further clarified. As authors mentioned that the magnitude of |A| from L1 norm might be detrimental for the prediction. What is the disagreement between L1 penalty and prediction quality? Why not apply these weights to L1 norm as a weighted L1 norm to control the scaling of A?
4. Authors stated that the last step is to build a mapping from the GPS features into the response Y. They mentioned that linear fully connected layer or a more complex neural network can be build on top of the GPS features. However, no detailed information is given in the paper. In the experiments, authors only stated that “we fit the GPS architecture using UPS optimizer for varying degree of the neighborhood of the graph”, and then the graph is used to train existing models as the input of the graph. Which architecture is used for building the mapping ?

In the experimental results, detailed definition or explanation of the compared methods and different settings should be clarified. For example, what is GPS 8, GCN_2 Eq. 9 in Table 1, and GCN_3 9 and GPS_1, GPS_2, GPS_3 and so on. More explanations of Figure 2 and the visualization method can be great helpful to understand the advantages of the proposed algorithm.

---

> ### Author Response · Authors · 2018-01-05
> **Thank you for the review.**
>
> Thank you for your comments. Below are the answers to some of your question
>
> --> 1. sigma is never clarified in the main context or experiments
>
> Sigma is a ReLU in our experiments.
>
> --> 2. the shared weights should be relevant to the ordering of neighbors, instead of the set of neighbors without ordering, in which case, the sharing looks random.
>
> Ordering of neighbors is fixed to be in alignment with the order of vertices in the graph adjacency matrix. Computationally, this is easily achievable by taking the inner product of a row of a graph adjacency matrix and a weight vector (weight vector is shared across rows).
>
> --> 3. another explanation about the weights as the rescaling to matrix A needs to further clarified. As authors mentioned that the magnitude of |A| from L1 norm might be detrimental for the prediction. What is the disagreement between L1 penalty and prediction quality? Why not apply these weights to L1 norm as a weighted L1 norm to control the scaling of A?
>
> L1 penalty acts as a regularizer, so the optimization approach will favor a sparser graph at a cost of sacrificing some of the performance. Same phenomena is observed in linear regression - coefficients learned with LASSO penalty are biased and refitting the regression with only selected variables generally improves the predictive performance. If we apply weights to the L1 norm of the graph adjacency and try to optimize for these weights, it will blow up the objective function ("optimal" weights will go to minus infinity).
>
> --> 4. Authors stated that the last step is to build a mapping from the GPS features into the response Y. They mentioned that linear fully connected layer or a more complex neural network can be build on top of the GPS features. However, no detailed information is given in the paper. In the experiments, authors only stated that “we fit the GPS architecture using UPS optimizer for varying degree of the neighborhood of the graph”, and then the graph is used to train existing models as the input of the graph. Which architecture is used for building the mapping ?
>
> We used linear mapping from GPS features to Y in the experiments.
>
> --> In the experimental results, detailed definition or explanation of the compared methods and different settings should be clarified. For example, what is GPS 8, GCN_2 Eq. 9 in Table 1, and GCN_3 9 and GPS_1, GPS_2, GPS_3 and so on. More explanations of Figure 2 and the visualization method can be great helpful to understand the advantages of the proposed algorithm.
>
> In GPS 8 and GCN 9, 8 and 9 are the corresponding equation references. Subscript numbers correspond to the number of layers for the GCN and maximum degree of the adjacency matrix polynomial for the GPS. We will improve the clarity of the experimental section and add the requested details.

---

### Official Review · AnonReviewer1 · 2017-11-26
**rejection**

**Rating:** 3
**Confidence:** 3

**Review:**

There are many language issues rendering the text hard to understand, e.g.,
-- in the abstract: "several convolution on graphs architectures"
-- in the definitions: "Let data with N observation" (no verb, no plural, etc).
-- in the computational section: "Training size is 9924 and testing is 6695. "
so part of my negative impression may be pure mis-understanding of what
the authors had to say.

Still, the authors clearly utilise basic concepts (c.f. "utilize eigenvector
basis of the graph Laplacian to do filtering in the Fourier domain") in ways
that do not seem to have any sensible interpretation whatsoever, even allowing
for the mis-understanding due to grammar. There are no clear insight,
no theorems, and an empirical evaluation on an ill-defined problem in
time-series forecasting. (How does it relate to graphs? What is the graph
in the time series or among the multiple time series? How do the authors
implement the other graph-related approaches in this problem featuring
time series?) My impression is hence that the only possible outcome is

rejection.

---

> ### Author Response · Authors · 2018-01-05
> **Thank you for the review.**
>
> Thank you for your comments. We will improve the writing of the paper. Below are the answers to some of your questions.
>
> --> Still, the authors clearly utilise basic concepts (c.f. "utilize eigenvector basis of the graph Laplacian to do filtering in the Fourier domain") in ways that do not seem to have any sensible interpretation whatsoever, even allowing for the mis-understanding due to grammar
>
> Graph Laplacian is a symmetric matrix and therefore it has orthonormal eigenvectors. Collection of orthonormal vectors forms a basis. Hence, eigenvector basis is a set of eigenvectors. Filtering on a graph can be performed in the spectral (Fourier) domain via Eq. (10) in the paper. Note that the formula involves eigenvectors (i.e. eigenvector basis) of the graph Laplacian.
>
>
> --> There are no clear insight
>
> We showed that it is possible to learn a graph by building neural network differentiable with respect to the graph adjacency matrix. Next, we analyzed importance of the input graph in various settings and found that for some of the recently proposed architectures input graph does not make any noticeable difference (i.e. random, kNN and graph learned via UPS all resulted in similar performance in the cases of ChebNet and ConvNet). This result seems a bit worrisome as one would not expect to see good performance with a random graph when a neural network is built to utilize the graph.
>
>
> --> Empirical evaluation on an ill-defined problem in time-series forecasting. (How does it relate to graphs? What is the graph in the time series or among the multiple time series? How do the authors implement the other graph-related approaches in this problem featuring time series?)
>
> Consider an example: we observe wind and precipitation measurements from various weather stations. Observations from a weather station can be used to predict tomorrow’s weather at another, spatially close, weather station. In this example graph can be constructed based on additional spatial information. In the application we consider such additional information is not available and hence learning the graph is important.

---

### Official Review · AnonReviewer3 · 2017-12-11
**Added Late Review**

**Rating:** 4
**Confidence:** 3

**Review:**

The authors develop a novel scheme for backpropagating on the adjacency matrix of a neural network graph.  Using this scheme, they are able to provide a little bit of evidence that their scheme allows for higher test accuracy when learning a new graph structure on a couple different example problems.

Pros:
-Authors provide some empirical evidence for the benefits of using their technique.
-Authors are fairly upfront about how, overall, it seems their technique isn't doing *too* much--null results are still results, and it would be interesting to better understand *why* learning a better graph for these networks doesn't help very much.

Cons:
-The grammar in the paper is pretty bad.  It could use a couple more passes with an editor.
-For a, more or less, entirely empirical paper, the choices of experiments are...somewhat befuddling.  Considerably more details on implementation, training time/test time, and even just *more* experiment domains would do this paper a tremendous amount of good.
-While I mentioned it as a pro, it also seems to be that this technique simply doesn't buy you very much as a practitioner.  If this is true--that learning better graph representations really doesn't help very much, that would be good to know, and publishable, but actually *establishing* that requires considerably more experiments.

Ultimately, I will have to suggest rejection, unless the authors considerably beef up their manuscript with more experiments, more details, and improve the grammar considerably.

---

> ### Author Response · Authors · 2018-01-05
> **Thank you for the review.**
>
> Thank you for you comments.
>
> --> If this is true--that learning better graph representations really doesn't help very much, that would be good to know, and publishable, but actually *establishing* that requires considerably more experiments.
>
> We agree with your opinion and are working on additional simulated and real data experiments to investigate the observed phenomena. We summarized the findings in our general response comment.
>
> We will revise the manuscript to improve the grammar.

---

### Author Response · Authors · 2018-01-05
**Author rebuttal**

We thank all the reviewers for their comments and questions. Individual responses are provided as comments to your reviews. Unfortunately, we have not yet finished the revision of the manuscript as we are working on a more methodological way to assess the role of the input graph in the cases of ChebNet and ConvNet. In the case of 20news groups data, we have experimented with different node degree distributions for generating a random graph (at fixed sparsity level). Additionally, we considered an extreme case of a graph with a randomly chosen subset of vertices forming a fully connected component, while all other vertices are mutually disconnected. We observed that non of that (even the extreme scenario) altered the behavior of neither ChebNet nor ConvNet. This can be explained by the usage of too many learnable parameters in the ChebNet and ConvNet architectures, making it possible for them to adjust to any input graph in high-dimensional setting. We are currently working on exploring a lower dimensional scenario via simulation experiments. GPS architecture (with a fixed graph) performed noticeably worse in the extreme scenario of random graph generation.

---

### Decision · Program_Chairs · 2018-01-29
**ICLR 2018 Conference Acceptance Decision**

**Decision:**

Reject

**Comment:**

This paper addresses the problem of learning neural graph representations, based on graph filtering techniques in the vertex domain.

Reviewers agreed on the fact that this paper has limited interest in its current form, and has serious grammatical issues. The AC thus recommends rejection at this time.